# The Causal Relationship Between Choline Metabolites and Acute Acalculous Cholecystitis: Identifying *ABCG8* as Colocalized Gene

**DOI:** 10.3390/nu16213588

**Published:** 2024-10-22

**Authors:** Yuntong Gao, Kun Mao, Congying Yang, Xisu Wang, Shixuan Liu, Zimeng Ma, Qi Zhai, Liang Shi, Qian Wu, Tianxiao Zhang

**Affiliations:** 1Department of Epidemiology and Biostatistics, School of Public Health, Xi’an Jiaotong University Health Science Center, Xi’an 710061, China; altogik2002@stu.xjtu.edu.cn (Y.G.); submacar@stu.xjtu.edu.cn (K.M.); yangcy022@stu.xjtu.edu.cn (C.Y.); doofzijing@outlook.com (X.W.); liushixuan2022@gmail.com (S.L.); mzmbla0321@163.com (Z.M.); zhaiqi0722@163.com (Q.Z.); shi-liang@stu.xjtu.edu.cn (L.S.); 2National Anti-Drug Laboratory, Shaanxi Regional Center, Xi’an 712000, China

**Keywords:** Mendelian randomization, choline metabolites, acute acalculous cholecystitis, low-density lipoprotein, *ABCG8*

## Abstract

Background: Acute acalculous cholecystitis (AAC) is a type of cholecystitis with high mortality rate while its pathogenesis remains complex. Choline is one of the essential nutrients and is related to several diseases. This study aimed to explore the causal relationship between choline metabolites and AAC and its potential mechanisms. Methods: This research utilized the two-sample Mendelian randomization method to investigate the causal relationship between choline metabolites and AAC. Additionally, multivariable Mendelian randomization and mediated Mendelian randomization were used to explore potential confounding effects from low-density lipoprotein (LDL), high-density lipoprotein (HDL), triglycerides (TGs), and coronary artery disease (CAD). Linkage disequilibrium score regression (LDSC), co-localization analysis, and enrichment analysis were used to investigate relevant molecular mechanisms. Results: There is a negative causal relationship between total choline (OR [95%CI] = 0.9982 [0.9974, 0.9990], *p* = 0.0023), phosphatidylcholine (OR [95%CI] = 0.9983 [0.9976–0.9991], *p* = 0.0040), sphingomyelin (OR [95%CI] = 0.9980 [0.9971–0.9988], *p* = 0.0001), and AAC. The mediating effects of LDL were −0.0006 for total choline, −0.0006 for phosphatidylcholine, and −0.0008 for sphingomyelin, indicating a protective effect of total choline, phosphatidylcholine, and sphingomyelin on AAC. Colocalized SNP rs75331444, which is mapped to gene *ABCG8*, was identified for total choline (PPH4 = 0.8778) and sphingomyelin (PPH4 = 0.9344). Conclusions: There is a causal relationship between choline metabolites and cholecystitis, mediated through the protective action of LDL. Our results suggest that *ABCG8* may play a role in the development of non-calculous cholecystitis.

## 1. Introduction

Cholecystitis is one of the most common digestive diseases occurring as a complication with cholelithiasis, affecting 10% of the global population [1]. Acute cholecystitis may occur with gallstones or without gallstones [2]. The latter case is called acute acalculous cholecystitis (AAC), accounting for 2–15% of all cases of acute cholecystitis [3]. The pathogenesis of AAC is complex. Ischemia–reperfusion injury, along with the involvement of proinflammatory eicosanoid mediators, is considered the core of this process [4]. Cholestasis, opioid therapy, positive-pressure ventilation, and total parenteral nutrition have also been considered contributing factors [5]. AAC is characterized by a significant mortality rate and is related to various complications [6]. Therefore, further research is needed on the pathogenic mechanisms of AAC.

Choline is an essential human nutrient involved in a variety of biological functions, including neurotransmission, membrane synthesis, lipid transport, and one-carbon metabolism [7]. It plays an important role in human growth and development and has been negatively associated with a variety of diseases, including birth defects, neurodevelopmental and cognitive alterations, hepatic steatosis, cardiovascular disease (CVD), and cancer [7]. Choline exists in several forms, including water-soluble free choline, fat-soluble phosphatidylcholine, and sphingomyelin [8]. A protective causal relationship between sphingomyelin and cholelithiasis was suggested in previous studies [9], while the relationship between choline and its metabolites and AAC has not been established.

Mendelian randomization is a research approach that uses randomly assigned genetic variants as instrumental variables to infer causal relationships between exposures and outcomes [10]. In addition, co-localization is an essential analytical method for exploring the common causal molecular mechanism between different phenotypes [11]. In this study, we aimed to explore the underlying causal relationship between choline and its metabolites and cholecystitis in vivo through Mendelian randomization analyses, and to further test the genetic correlation and identify colocalized genes.

## 2. Materials and Methods

### 2.1. Study Design

Mendelian randomization (MR) is a method used in epidemiology and genetics to investigate the causal relationship between an exposure (such as a risk factor or treatment) and an outcome (such as a disease). The key idea of MR is to identify genetic variants (typically single nucleotide polymorphisms [SNPs]) that serve as instrumental variables (IVs), and these variants need to meet three conditions: (1) the variant is associated with the exposure; (2) the variant is not associated with the outcome via a confounding pathway; and (3) the variant does not directly affect the outcome, only possibly indirectly via the exposure [12]. By examining the association between the selected DNA variants and the outcome, the potential causal relationship between the exposures and the outcome is determined [13]. In the present study, we implemented two-sample Mendelian randomization (TSMR) of which the relationships of IVs with exposures and outcomes were measured from two samples [14].

### 2.2. Data Sources

We took three choline metabolites as our exposure factors from UK Biobank (n = 114,999), including total choline, phosphatidylcholine, and sphingomyelin, and we downloaded the data through the IEU Open GWAS database (https://gwas.mrcieu.ac.uk) (10 November 2023) [15]. SNP-Cholecystitis (ICD-10 code K81) data were taken from the Neale lab in the UK Biobank (UKB) (http://www.nealelab.is/uk-biobank) (10 November 2023). In the multivariable MR analysis, we extracted data for four additional variables: low-density lipoprotein (LDL): ieu-b-5089 (n = 201,678); high-density lipoprotein (HDL): ieu-b-109 (n = 403,943) [16]; triglycerides (TGs): ieu-b-111 (n = 441,016) [16]; and coronary vascular disease (CAD): ebi-a-GCST90013868 (n = 352,063) [17]. Multiple inter-institutional selection was attempted to avoid sample overlap between exposures and outcomes, and all samples used in this study were obtained from the European population. These published data have undergone quality control and ethical review and are ready for immediate use.

### 2.3. Two-Sample Mendelian Randomization Analysis

We selected SNPs with strong correlations with exposures and outcomes at the *p* < 5 × 10^−8^ level and further removed variants in linkage disequilibrium (r^2^ < 0.001 over a scan window of 10,000 kb). In addition, F-tests were performed and SNPs with F-static values <10 were excluded. Finally, we identified 49 SNPs for total choline, 48 SNPs for phosphatidylcholine, and 48 SNPs for sphingomyelin, which were used in subsequent Mendelian randomization analyses.

Five methods including MR Egger [18], weighted median [19], inverse-variance-weighted (IVW) [20], simple mode, and weighted mode [21] were utilized to model the causal relationship between exposures and outcomes. A significant causal relationship is considered when the *p*-value of IVW is <0.05 and the results of the five methods have the same direction. The results were tested for heterogeneity using the Cochran Q test [22]. The MR-Egger intercept [23] and the MR presso global test [24] were used to test for pleiotropy. If heterogeneity and/or pleiotropy were detected, the MR presso outlier method [25] was implemented to remove outliers. Sensitive analysis was performed based on the IVW results through leave-one-out tests for each SNP [25]. In addition, to control the potential confounding effects, multivariable Mendelian randomization analyses were performed. LDL, HDL, TGs, and CAD were included as covariates. The IVW random-effects model and MR-Egger model were utilized for TSMR analysis.

### 2.4. Reverse and Mediated Mendelian Randomization Analysis

To further elucidate the causal role of LDL, we conducted mediated Mendelian randomization analysis with LDL as the mediating factor. Initially, we examined the reverse Mendelian randomization results; total choline, phosphatidylcholine, and sphingomyelin were selected as the outcomes; and cholecystitis was selected as the exposure factor. In the mediator of Mendelian randomization analysis, we first employed a two-sample Mendelian randomization approach to estimate the effect of total choline, phosphatidylcholine, and sphingomyelin on LDL (*β*_1_). Subsequently, using LDL as the exposure and cholecystitis as the outcome, we estimated the direct effect of the mediator on cholelithiasis (*β*_2_). The indirect effect of sphingomyelin on cholelithiasis through the mediator was calculated using *β*_1_ × *β*_2_, and its significance was tested using a stepwise testing method.

### 2.5. Linkage Disequilibrium Score Regression (LDSC) and Co-Localization Analysis

The LDSC analysis was performed to investigate the genetic correlation between three choline metabolites and AAC [26]. Co-localization analysis was implemented using the coloc package to further examine the potential shared causal variants between the three choline metabolites and AAC [13].

### 2.6. Gene Enrichment Analysis

The gene list was extracted based on SNPs identified from co-localization analysis. The STRING database (https://string-db.org) (22 November 2023) was utilized to obtain a set of genes which are based on protein–protein interaction (PPI) data [27]. Gene set enrichment analysis was then performed for this gene set using GO and KEGG databases [28].

## 3. Results

### 3.1. Causal Relationship between Choline Metabolites and AAC

The analysis pipeline of the present study is shown in Figure 1. A significant causal relationship was identified between the three choline metabolites and AAC. The IVW results indicated a significant negative causal relationship between total choline (OR [95%CI] = 0.9982 [0.9974, 0.9990], *p* = 0.0023), phosphatidylcholine (OR [95%CI] = 0.9983 [0.9976–0.9991], *p* = 0.0040), sphingomyelin (OR [95%CI] = 0.9980 [0.9971–0.9988], *p* = 0.0001), and AAC (Figure 2). Neither significant heterogeneity (Cochran Q test, *p* > 0.05) nor pleiotropy (MR-Egger global test, *p* > 0.05) was detected (Appendix A). The results of leave-one-out tests and funnel plots are summarized in Appendix A. No individual SNPs were identified to dictate the IVW results. The funnel plots were approximately symmetrical. To further validate the causal relationship identified from TSMR, multivariable Mendelian randomization analyses were further performed. The serum levels of HDL, LDL, TGs, and CAD were adjusted and the results are summarized in Table 1. Interestingly, the causal relationship between all the three choline metabolites and AAC did not remain significant after being adjusted for the serum level of LDL (total choline: OR [95%CI] = 0.9985 [0.9978,0.9991], *p* = 0.1153); phosphatidylcholine (OR [95%CI] = 0.9989 [0.9984,0.9994], *p* = 0.2219); or sphingomyelin (OR [95%CI] = 0.9983 [0.9976,0.9991], *p* = 0.1660). Significant horizontal pleiotropy was identified using the MR-Egger intercept in the multivariable models adjusting for HDL and TGs. The overall F value for the selected instrumental variables was greater than 10 in all groups (Appendix A). 

### 3.2. Reverse and Mediator Mendelian Randomization Analysis

The IVW results of the reverse Mendelian randomization analysis revealed no significant causal relationship between cholecystitis and total choline (*p* = 0.2520), phosphatidylcholine (*p* = 0.3424), or sphingomyelin (*p* = 0.1100). When LDL was considered as the outcome, significant positive causal relationships were observed with total choline (*β*_1_ = 0.3533, *p* = 1.66 × 10^−16^), phosphatidylcholine (*β*_1_ = 0.3468, *p* = 2.17 × 10^−28^), and sphingomyelin (*β*_1_ = 0.4682, *p* = 1.31 × 10^−51^). Additionally, a significant inverse causal relationship was found when LDL was the exposure and cholecystitis was the outcome (*β*_2_ = −0.0018, *p* = 0.0022). The mediating effects of LDL were −0.0006 for total choline, −0.0006 for phosphatidylcholine, and −0.0008 for sphingomyelin. A causal steps approach confirmed the significance of LDL’s mediating effect in the causal relationships between total choline (*p* = 0.0041), phosphatidylcholine (*p* = 0.0031), and sphingomyelin (*p* = 0.0027) with cholecystitis.

### 3.3. Genetic Correlation between the Three Choline Metabolites and AAC

Although no genome-wide genetic correlations were detected between the three choline metabolites and AAC (Table 2), a significant colocalized SNP, rs75331444, was identified for total cholines (PPH4 = 0.8778) and sphingomyelin (PPH4 = 0.9344) (Figure 3). This SNP is mapped to gene *ABCG8*. No colocalization signal was detected in the colocalization analysis of phosphatidylcholine and AAC (Appendix A).

### 3.4. Enrichment Pathway for ABCG8

The colocalized gene *ABCG8* was found to be enriched in multiple biological processes, cellular components, and molecular functions in the GO database (Figure 4A). In the KEGG database, this gene was identified to be enriched in bile secretion, cholesterol metabolism, the thyroid hormone signaling pathway, ABC transporters, and proximal tubule bicarbonate reclamation (Figure 4B).

## 4. Discussion

Choline, a trace component of plasma, is involved in various key physiological functions in the body and drives disease progression. The causal relationship between plasma levels of total choline, phosphatidylcholine, and sphingomyelin and cholecystitis was evaluated utilizing Mendelian randomization, LDSC, co-localization analysis, and enrichment analysis. The two-sample Mendelian randomization results indicated that total choline, phosphatidylcholine, and sphingomyelin had a protective effect on AAC.

Multivariable Mendelian randomization was applied to assess the reliability of two-sample Mendelian randomization analysis and explore the sources of causality. In the multivariable Mendelian randomization results, it was observed that all positive results became negative after adjusting LDL. This suggested that LDL is prone to be an important confounder in this process, which was not seen with other possible confounders (HDL, triglycerides, and CAD). The results of mediated Mendelian randomization suggest that high levels of LDL are a protective factor for AAC, and choline may enhance this effect. This also explains the non-significant results in multivariable analysis after adjusting for LDL levels. Previous study has found a minor protective effect of serum low-density lipoprotein (LDL) cholesterol on cholecystitis, which is partly in accordance with this study. [29] LDL has been shown to interact with the immune system in several studies. [30] It has been shown that LDL can reduce lipopolysaccharide mediated central and peripheral inflammation [31] and bacterial infection may be one of the etiologies of AAC, but its specific role in cholecystitis still lacks specialized research support. Alternatively, LDL is involved in bile acid metabolism [32], which is responsible for the transport of cho-lesterol outward into the liver tissue, and cholestasis is one of the etiologies of AAC.

Moreover, choline retains its independent protective effect, potentially linked to the expression levels of certain proteins. To explore the source of the causal relationship between choline and cholecystitis, LDSC and colocalization were used. In LDSC analysis, there were co-localization sites between total choline, sphingomyelin, and cholecystitis. The results of the colocalization analysis suggested that there may be a particular driver molecule between total choline, sphingomyelin, and cholecystitis. Sphingomyelin accounts for a high proportion of total choline in the body [33]. In our results, both co-localization analyses had the same site, and the co-localization intensity of total choline was lower than that of sphingolipids. This suggested that the positive co-localization result of total choline with cholecystitis may be due to sphingolipids.

*ABCG8* was localized through colocalization analysis. It may play an important role in the causal relationship between choline metabolites and cholecystitis. *ABCG8* is exclusively expressed in hepatocytes, gallbladder epithelial cells, and intestinal cells, where it interacts with the ATP-binding cassette transporter G5 (ABCG5), which forms a specialized heterodimer involved in sterol metabolism [34]. Common mutations in *ABCG8* confer most of the genetic risk for cholelithiasis, accounting for approximately 25% of the total risk [35]. Previous studies have shown that *ABCG8* expression is elevated in patients with cholesterol gallstone disease and cholecystitis [36], and our study further demonstrated that *ABCG8* also plays a role in the development of non-calculous cholecystitis. This suggests that the molecular mechanisms between cholelithiasis and non-calculous cholecystitis have some similarities. Previous studies have demonstrated the role of ABCG8 in excretion of cholesterol [37,38]. Taken together, choline metabolites may play a protective role against AAC by promoting cholesterol excretion and transport. The specific molecular mechanism still needs further animal experiments. Cholestasis is one of the possible pathogenic factors of AAC. This strengthens the causal relationship between choline and AAC. Current animal studies have focused on choline with liver [39,40]. There are few animal studies on the relationship between choline and ACC. Animal experiments are needed in the future.

This research also has some limitations. Our study focused on the European population, making it difficult to extrapolate the findings to other populations. We did not analyze all choline metabolites, due to missing data, which may bring a new perspective to this study. The current research reveals causal reference, and wouldn’t have great significance for clinical practice. Related clinical translational research is needed in the future.

## 5. Conclusions

In summary, our study, for the first time, confirms a causal relationship between choline metabolites and cholecystitis, mediated through the protective action of LDL, and identifies the possible loci responsible for this causal relationship, suggesting a role for *ABCG8* in the development of non-calculous cholecystitis, which provides valuable information for probing its molecular mechanisms. Moreover, this finding could provide significant insights for individuals regarding their nutritional intake, especially the parenterally nourished patients.

## Figures and Tables

**Figure 1 nutrients-16-03588-f001:**
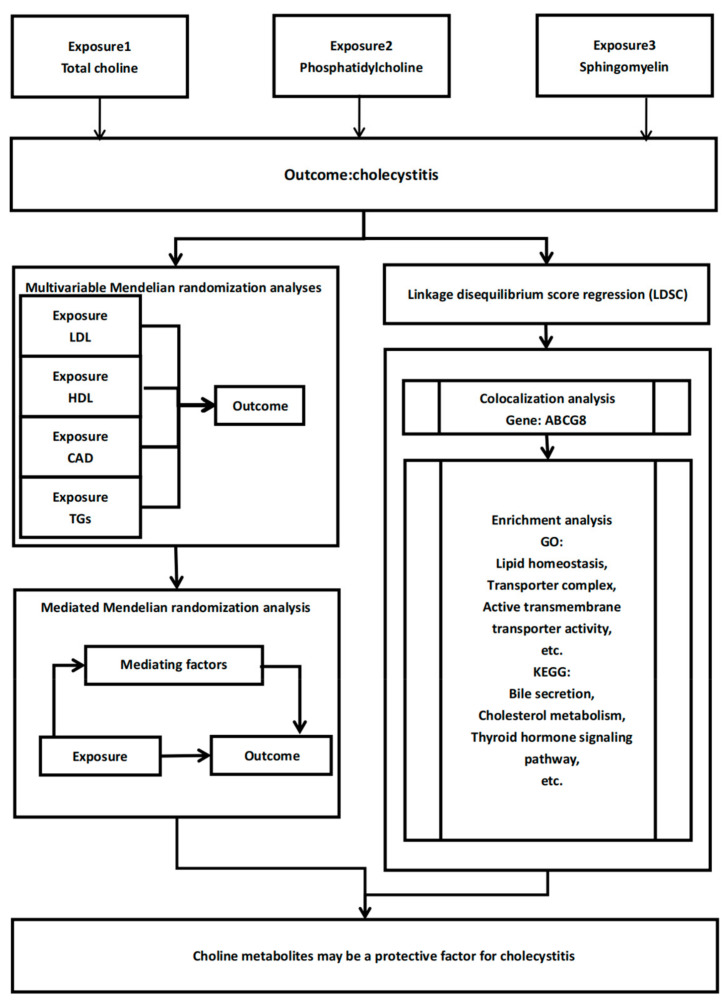
Conceptual framework of this study.

**Figure 2 nutrients-16-03588-f002:**
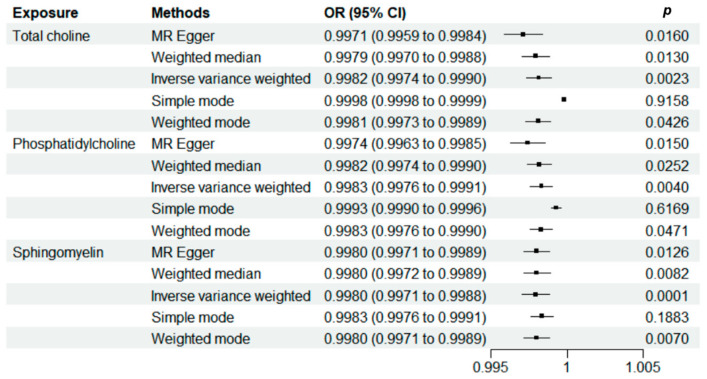
Analysis of the relationship between choline metabolites and cholecystitis by Mendelian randomization analysis (results corresponding to five different methods).

**Figure 3 nutrients-16-03588-f003:**
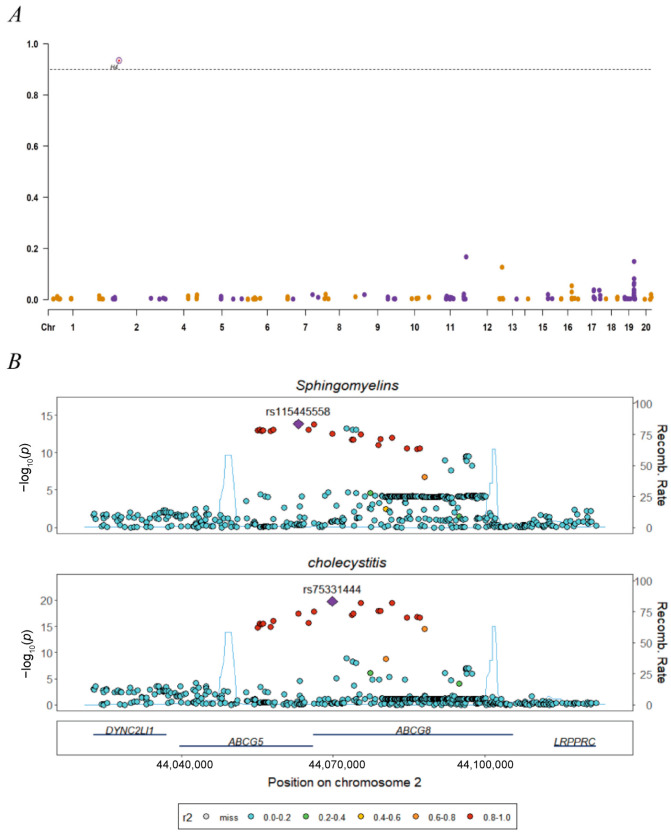
Manhattan plots for colocalization analysis. (**A**) Manhattan plot of selected SNP associations with PPH4 at the genome-wide scale with yellow and purple dots to distinguish between adjacent chromosomal locations. (**B**) Locus comparison plot for the COLOC analysis of the notable colocalization regions (PPH4 > 0.9).

**Figure 4 nutrients-16-03588-f004:**
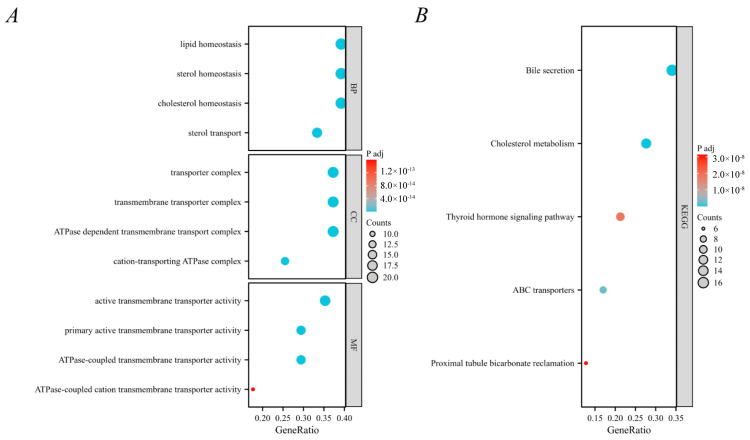
Bubble plots for enrichment analysis of genes and their related genes obtained from colocalization analysis. (**A**) GO enrichment analysis, including biological processes (BPs), cellular components (CCs), and molecular functions (MFs). (**B**) KEGG enrichment analysis.

**Table 1 nutrients-16-03588-t001:** Multivariable Mendelian randomization analysis between choline metabolites and cholecystitis.

Exposure	nSNPs	MVMR-IVW		MVMR-Egger		*p* for MR-Egger Intercept	F Statistic
		OR (95%CI)	*p*	OR (95%CI)	*p*		
HDL	321	0.9994 (0.9992 to 0.9997)	0.3446	1.0014 (1.0008 to 1.0020)	0.0916	0.0003	17.3554
Total choline		0.9977 (0.9967 to 0.9987)	0.0051	0.9975 (0.9965 to 0.9986)	0.0022		11.1154
LDL	90	0.9984 (0.9976 to 0.9991)	0.0618	0.9983 (0.9975 to 0.9990)	0.0895	0.867	52.8175
Total choline		0.9985 (0.9978 to 0.9991)	0.1153	0.9984 (0.9977 to 0.9991)	0.1243		47.1951
Triglyceride	278	1.0005 (1.0003 to 1.0008)	0.3097	0.9994 (0.9991 to 0.9997)	0.4261	0.0393	114.9307
Total choline		0.9984 (0.9978 to 0.9991)	0.0192	0.9984 (0.9977 to 0.9991)	0.0137		22.0579
CAD	88	0.9996 (0.9994 to 0.9998)	0.263	0.9997 (0.9995 to 0.9998)	0.5529	0.8148	29.6231
Total choline		0.9979 (0.9969 to 0.9988)	0.0001	0.9979 (0.9970 to 0.9988)	0.0001		82.2872
HDL	322	0.9992 (0.9989 to 0.9996)	0.2120	1.0013 (1.0007 to 1.0018)	0.1218	0.0002	19.9792
Phosphatidylcholine		0.9981 (0.9973 to 0.9989)	0.0140	0.9979 (0.9970 to 0.9988)	0.0006		12.8171
LDL	91	0.9981 (0.9973 to 0.9989)	0.0299	0.9979 (0.9970 to 0.9988)	0.0440	0.7143	68.9316
Phosphatidylcholine		0.9989 (0.9984 to 0.9994)	0.2219	0.9988 (0.9983 to 0.9993)	0.2040		58.3947
Triglyceride	280	1.0006 (1.0003 to 1.0008)	0.2737	0.9995 (0.9993 to 0.9997)	0.5045	0.0480	107.0640
Phosphatidylcholine		0.9986 (0.9979 to 0.9992)	0.0229	0.9985 (0.9979 to 0.9992)	0.0185		24.7822
CAD	86	0.9996 (0.9994 to 0.9997)	0.2560	0.9997 (0.9996 to 0.9999)	0.6550	0.6491	29.8555
Phosphatidylcholine		0.9982 (0.9974 to 0.9990)	0.0003	0.9982 (0.9975 to 0.9990)	0.0007		91.7388
HDL	316	0.9993 (0.9990 to 0.9996)	0.2730	1.0014 (1.0008 to 1.0021)	0.0929	0.0002	29.3628
Sphingomyelin		0.9978 (0.9968 to 0.9987)	0.0029	0.9978 (0.9968 to 0.9988)	0.0026		16.1736
LDL	90	0.9989 (0.9985 to 0.9994)	0.3740	0.9988 (0.9983 to 0.9993)	0.3520	0.7679	23.9513
Sphingomyelin		0.9983 (0.9976 to 0.9991)	0.1660	0.9983 (0.9975 to 0.9990)	0.1590		24.7277
Triglyceride	277	1.0000 (1.0000 to 1.0000)	0.9573	0.9989 (0.9984 to 0.9994)	0.1858	0.0829	77.8575
Sphingomyelin		0.9977 (0.9967 to 0.9987)	0.0005	0.9975 (0.9965 to 0.9986)	0.0002		26.9886
CAD	91	0.9997 (0.9996 to 0.9998)	0.528	1.0006 (1.0003 to 1.0009)	0.3730	0.0679	28.0342
Sphingomyelin		0.9976 (0.9965 to 0.9986)	0.0001	0.9977 (0.9967 to 0.9987)	0.0002		83.5995

MVMR-Egger: multivariable Mendelian randomization using Egger regression; MVMR-IVW: multivariable Mendelian randomization using inverse-variance-weighted approach; nSNPs: number of SNPs used in MR; LDL: low-density lipoprotein; HDL: high-density lipoprotein; CAD: coronary artery disease. The grayish-white background shows where the adjust variable is applied and makes the table intuitive.

**Table 2 nutrients-16-03588-t002:** Genetic correlation between choline metabolites and cholecystitis.

Exposure	Outcome	Rg	*p*
Total choline	Cholecystitis	−0.0940	0.6372
Phosphatidylcholine	Cholecystitis	−0.1077	0.5836
Sphingomyelin	Cholecystitis	−0.2796	0.1645

Rg: genetic correlation.

## Data Availability

The original data presented in this study are openly available in UK Biobank (https://www.ukbiobank.ac.uk/) (10 November 2023)—low-density lipoprotein (LDL): ieu-b-5089 (n = 201,678); high-density lipoprotein (HDL): 6ieu-b-109 (n = 403,943) [16]; triglycerides (TGs): ieu-b-111 (n = 441,016) [16]; and coronary vascular disease (CAD): ebi-a-GCST90013868 (n = 352,063) [17].

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
