# Peer review of "The Causal Relationship Between Choline Metabolites and Acute Acalculous Cholecystitis: Identifying ABCG8 as Colocalized Gene"

_nutrients, 2024, doi:10.3390/nu16213588_

Round 1
Reviewer 1 Report
Comments and Suggestions for Authors
This paper examines the relationship between choline metabolites and acute acalculous cholecystitis, and we believe it is an appropriate study with Informed Consent and Research Ethics Committee approval. The quality of the data and the argument are fine, and the paper is well thought out. There is nothing that requires improvement in this regard.
I will mention a few points to point out, but they require consideration in the discussion and introduction.
1. You state that the etiology of AAC (acute acalculous cholecystitis) is unknown, but I would like to see more description of what specific condition is hypothesized. Is there a small possibility of other etiologies?
2. You state that the purpose of this study is to "elucidate the causal relationship between choline metabolites and AAC." Please describe in detail the purpose of this study, specifically what hypothesis you wish to test. You should also state more clearly your hypothesis on how choline metabolites are involved in AAC and the possibilities that can be inferred from the results revealed in this study.
3. A protective effect via LDL (low-density lipoprotein) is described, but the specific mechanism why LDL has this protective effect is not well explained. Supplementing this point would add depth to the interpretation of the results.
4. Statistical results (ORs and P-values) are presented, but it is unclear how clinically significant these numbers are. For example, an odds ratio of 0.9982 indicates a very slight effect, but the clinical significance of this effect needs to be discussed.
5. A relationship between the ABCG8 gene and AAC has been noted, but the specific mechanism or hypothesis of how this gene is involved in the pathogenesis of AAC is unclear. A more detailed explanation based on more detailed gene function and previous studies is needed.
Author Response
- You state that the etiology of AAC (acute acalculous cholecystitis) is unknown, but I would like to see more description of what specific condition is hypothesized. Is there a small possibility of other etiologies?
Response: We apologize for the lack of clarification. According to previous research, gallbladder ischemia-reperfusion injury, cholestasis, bacterial infections, and abnormal anatomical structure of the biliary tract system could be possible etiologies. While the pathogenesis of AAC is still not fully understood, and it is often regarded as a relatively independent clinical disease[1]. We have added a couple sentences in to the abstract and introduction parts of the revised manuscript as followed,
Abstract: Acute acalculous cholecystitis (AAC) is a type of cholecystitis with high mortality rate while its pathogenesis remains unclear complex.
Introduction: The pathogenesis of AAC is complex. Ischemia-reperfusion injury, along with the involvement of proinflammatory eicosanoid mediators, is considered to be the core of this process [2]. Cholestasis, opioid therapy, positive pressure ventilation, and total parenteral nutrition have also been considered as contributing factors [3].
- Fu Y, Pang L, Dai W, Wu S, Kong J (2022) Advances in the Study of Acute Acalculous Cholecystitis: A Comprehensive Review. Dig Dis 40:468-4781.
- Frazee, R.C., D.M. Nagorney, and P. Mucha, ACUTE ACALCULOUS CHOLECYSTITIS. Mayo Clinic Proceedings, 1989. 64(2): p. 163-167.
- Barie, P.S. and S.R. Eachempati, Acute acalculous cholecystitis. Current gastroenterology reports, 2003. 5(4): p. 302-9.
- You state that the purpose of this study is to "elucidate the causal relationship between choline metabolites and AAC." Please describe in detail the purpose of this study, specifically what hypothesis you wish to test. You should also state more clearly your hypothesis on how choline metabolites are involved in AAC and the possibilities that can be inferred from the results revealed in this study.
Response: We apologize for our lack of clarifications. Our hypothesis is that choline plays a role in the development and progression of AAC. To test this hypothesis, we performed causal validation using a Mendelian randomization approach and the results are positive. To further investigate how choline affects AAC, we have performed mediation Mendelian randomization analysis to test relevant possible mediation pathways. The results indicated that LDL might be a protective factor of ACC. In addition, the colocalization gene ABCG8 functions in the excretion of cholesterol. Taken together, we hypothesized that choline metabolites may play a protective role against AAC through promoting cholesterol excretion and transport. The specific molecular mechanism is unknown and further animal experiments are still needed.
3.A protective effect via LDL (low-density lipoprotein) is described, but the specific mechanism why LDL has this protective effect is not well explained. Supplementing this point would add depth to the interpretation of the results.
Response: We agree with the reviewer’s point that a specific mechanism should be stated and discussed to add depth to the interpretation of the results. Previous study has found a minor protective effect of serum low-density lipoprotein (LDL) cholesterol on cholecystitis, which is partly in accordance with this study[4]. LDL has been shown to interact with the immune system in several studies.[5] It has been shown that LDL can reduce lipopolysaccharide mediated central and peripheral inflammation and bacterial infection may be one of the etiologies of AAC[6], but its specific role in cholecystitis still lacks specialized research support. Alternatively, LDL is involved in bile acid metabolism[7], which is responsible for the transport of cholesterol outward into the liver tissue, and cholestasis is one of the etiologies of AAC. These may account for the protective effect of LDL. We have supplemented the discussion and cited references in this article as followed,
Previous study has found a minor protective effect of serum low-density lipoprotein (LDL) cholesterol on cholecystitis, which is partly in accordance with this study[4]. LDL has been shown to interact with the immune system in several studies. [5] It has been shown that LDL can reduce lipopolysaccharide mediated central and peripheral inflammation [6] and bacterial infection may be one of the etiologies of AAC, but its specific role in cholecystitis still lacks specialized research support. Alternatively, LDL is involved in bile acid metabolism [7], which is responsible for the transport of cholesterol outward into the liver tissue, and cholestasis is one of the etiologies of AAC. (Page 8 line 212-219)
- Yang H, Chen L, Liu K, Li C, Li H, Xiong K, Li Z, Lu C, Chen W, Liu Y (2021) Mendelian randomization rules out the causal relationship between serum lipids and cholecystitis. BMC Med Genomics 14:224
- Rhoads JP, Major AS (2018) How Oxidized Low-Density Lipoprotein Activates Inflammatory Responses. Crit Rev Immunol 38:333-342
- Radford-Smith DE, Yates AG, Rizvi L, Anthony DC, Probert F (2023) HDL and LDL have distinct, opposing effects on LPS-induced brain inflammation. Lipids in Health and Disease 22:54
- Sato R (2020) Recent advances in regulating cholesterol and bile acid metabolism. Bioscience, biotechnology, and biochemistry 84:2185-2192
- Statistical results (ORs and P-values) are presented, but it is unclear how clinically significant these numbers are. For example, an odds ratio of 0.9982 indicates a very slight effect, but the clinical significance of this effect needs to be discussed.
Response: We agree with the reviewer’s point that clinical significance of our results need to be fully discussed. In this study, the serum choline level, which is a continuous variable, were utilized as exposure factor. In this sense, the effect size reported would be quite moderate. The approach of Mendelian randomization is primarily designed to assess causality. The genetic effects derived from GWAS are population level effect, which cannot be applied to clinical practice. We have added the following texts in the discussion part of the revised manuscript,
The current research reveals causal reference, and wouldn’t have great significance for clinical practice. Related clinical translational research are needed in the future. (Page 9 line 251-252)
5.A relationship between the ABCG8 gene and AAC has been noted, but the specific mechanism or hypothesis of how this gene is involved in the pathogenesis of AAC is unclear. A more detailed explanation based on more detailed gene function and previous studies is needed.
Response: We apologize for our lack of clarifications. Previous studies have demonstrated the gene ABCG8 has played a role in excretion of cholesterol[8, 9]. On the other hand, cholestasis is one of the possible pathogenic factors of AAC. This strengthens the causal relationship between choline and AAC. We have added the following texts in the discussion part of the revised manuscript,
Previous studies have demonstrated the role of ABCG8 in excretion of cholesterol [8,9]. Taken together, choline metabolites may play a protective role against AAC by promoting cholesterol excretion and transport. The specific molecular mechanism still needs further animal experiments. Cholestasis is one of the possible pathogenic factors of AAC. This strengthens the causal relationship between choline and AAC. (Page 9 line 240-245)
- Coy DJ, Wooton-Kee CR, Yan B, Sabeva N, Su K, Graf G, Vore M (2010) ABCG5/ABCG8-independent biliary cholesterol ex-cretion in lactating rats. Am J Physiol Gastrointest Liver Physiol 299:G228-G235
- Yu X-H, Qian K, Jiang N, Zheng X-L, Cayabyab FS, Tang C-K (2014) ABCG5/ABCG8 in cholesterol excretion and athero-sclerosis. Clin Chim Acta 428:82-88
Reviewer 2 Report
Comments and Suggestions for Authors
The work presented by Gao et al. refers to a study that examines the relationship between choline metabolites and acute acalculous cholecystitis. The authors use extensive databases to perform two-sample Mendelian randomization. The work is overall well written and the experimental approach, to the best of my knowledge, judiciously applied.
Major comment: several conclusions are drawn from the study, establishing a causal relationship between choline metabolites and cholecystitis, mediated through a “putative” protective role of LDL. This LDL protective role is rather surprising and most certainly needs further consideration.
Comments on the Quality of English Language
Minor comments:
i) Page 2 line 81: “data from the Neale lab”; is this data from a published study? If that is the case why not including the reference?
ii) Page 3 line 94: word finally is repeated.
iii) Page 3 linee 106-107: “we used” and “were performed”; use one or the other, not the two expressions.
iv) Page 3 line 111: “examining”? correct word and the sentence needs also correction.
v) Page 4 line 4: “To further” and “were further”; no need to emphasize the further. Rewrite sentence.
vi) Page 5 line 171: “Casual” should be changed by Causal.
vii) Page 6 line 177: only two items are mentioned thus there is no need for the comma before “sphingomyelin”. Substitute comma by and!
viii) Page 8 line 199: “involves in”; needs correction, eventually substitute by is involved.
ix) Page 8 line 205: “randomization were”? needs correction.
Author Response
Major comment: several conclusions are drawn from the study, establishing a causal relationship between choline metabolites and cholecystitis, mediated through a “putative” protective role of LDL. This LDL protective role is rather surprising and most certainly needs further consideration.
Response: We agree with the reviewer’s point that the LDL protective role is a surprising finding and more discussion are needed to clarify it. Previous study has found a minor protective effect of serum low-density lipoprotein (LDL) cholesterol on cholecystitis, which is partly in accordance with this study.[1] LDL has been shown to interact with the immune system in several studies.[2] It has been shown that LDL can reduce lipopolysaccharide mediated central and peripheral inflammation[3] and bacterial infection may be one of the etiologies of AAC, but its specific role in cholecystitis still lacks specialized research support. Alternatively, LDL is involved in bile acid metabolism[4], which is responsible for the transport of cholesterol outward into the liver tissue, and cholestasis is one of the etiologies of AAC. These may account for the protective effect of LDL. The details still need to be verified by further animal experiments and the concrete functional pathway needs further consideration. We have added the following texts in the discussion part of the revised manuscript,
Previous study has found a minor protective effect of serum low-density lipoprotein (LDL) cholesterol on cholecystitis, which is partly in accordance with this study. [1] LDL has been shown to interact with the immune system in several studies. [2] It has been shown that LDL can reduce lipopolysaccharide mediated central and peripheral inflammation [3] and bacterial infection may be one of the etiologies of AAC, but its specific role in cholecystitis still lacks specialized research support. Alternatively, LDL is involved in bile acid metabolism [4], which is responsible for the transport of cho-lesterol outward into the liver tissue, and cholestasis is one of the etiologies of AAC. (Page 8 line 212-219)
- Yang H, Chen L, Liu K, Li C, Li H, Xiong K, Li Z, Lu C, Chen W, Liu Y (2021) Mendelian randomization rules out the causal relationship between serum lipids and cholecystitis. BMC Med Genomics 14:224
- Rhoads JP, Major AS (2018) How Oxidized Low-Density Lipoprotein Activates Inflammatory Responses. Crit Rev Immunol 38:333-342
- Radford-Smith DE, Yates AG, Rizvi L, Anthony DC, Probert F (2023) HDL and LDL have distinct, opposing effects on LPS-induced brain inflammation. Lipids in Health and Disease 22:54
- Sato R (2020) Recent advances in regulating cholesterol and bile acid metabolism. Bioscience, biotechnology, and biochemistry 84:2185-2192Minor comments
i) Page 2 line 81: “data from the Neale lab”; is this data from a published study? If that is the case why not including the reference?
Response: We apologize for our lack of clarifications. Data from the Neale lab wasn’t from a published study. The data was uploaded to the website: http://www.nealelab.is/uk-biobank/. According to the requirement of Neale lab, we cited the page in the manuscript.
ii) Page 3 line 94: word finally is repeated.
Response: We apologize for our negligence. We have revised the mistake in Page 3 line 94 as followed,
Page 3 line 94: Finally,
iii) Page 3 line 106-107: “we used” and “were performed”; use one or the other, not the two expressions.
Response: We apologize for our negligence. We have revised the mistake in Page 3 line 106-107 as followed,
In addition, to control the potential confounding effects, we used multivariable Mendelian randomization analyses were performed.
iv) Page 3 line 111: “examining”? correct word and the sentence needs also correction.
Response: We apologize for our negligence. We have revised the mistake in Page 3 line 111 as followed,
Initially, we examining examined the reverse Mendelian randomization results. total choline, phosphatidylcholine, and sphingomyelin were selected as the outcomes, and cholecystitis was selected as the exposure factor.
v) Page 4 line 4: “To further” and “were further”; no need to emphasize the further. Rewrite sentence.
Response: We apologize for our negligence. We have revised the mistake in Page 4 line 144,
To further validate the causal relationship identified from TSMR, we further performed multivariable Mendelian randomization analyses.
vi) Page 5 line 171: “Casual” should be changed by Causal.
Response: We apologize for our negligence. We have revised the mistake in Page 5 line 171 as followed,
Casual Causal steps approach confirmed the significance of LDL's mediating effect in the causal relationships between total choline (P = 0.0041), phosphatidylcholine (P = 0.0031), and sphingomyelin (P = 0.0027) with cholecystitis.
vii) Page 6 line 177: only two items are mentioned thus there is no need for the comma before “sphingomyelin”. Substitute comma by and!
Response: We apologize for our negligence. We have revised the mistake in Page 6 line 177 as followed,
Although no genome-wide genetic correlations were detected between the three choline metabolites and AAC (Table 2), a significant colocalized SNP, rs75331444 was identified for total cholines (PPH4 = 0.8778) and sphingomyelin (PPH4 = 0.9344) (Figure 3).
viii) Page 8 line 199: “involves in”; needs correction, eventually substitute by is involved.
Response: We apologize for our negligence. We have revised the mistake in Page 8 line 199 as followed,
Choline, a trace component of plasma, is involves involved in various key physiological functions in the body and drives disease progression.
ix) Page 8 line 205: “randomization were”? needs correction.
Response: We apologize for our negligence. We have revised the mistake in Page 8 line 205.
Multivariable Mendelian randomization were was applied to assess the reliability of two sample Mendelian randomization analysis and explore the sources of causality.
Reviewer 3 Report
Comments and Suggestions for Authors
I congratulate the authors on an interesting original work.
The manuscript fully meets all the criteria of the instructions for authors.
I would suggest the following minor changes of the manuscript:
1. If there are similar studies on animal models, please kindly mention them in your research, as well as supplement the references.
2. By searching the Medline database with the keyword "ABCG8", I see N=808 articles. I kindly ask you to supplement the references that are more important to you and expand the discussions with the relevant ones.
I have no other comments or complaints.
Best Regards and congratulations to the authors
Author Response
- If there are similar studies on animal models, please kindly mention them in your research, as well as supplement the references.
Response: We agree with the reviewer’s point that animal models studies should be mentioned in the manuscript. We have searched the literature related to the effects of choline. However, it seems that most of the current studies focused on the correlation between choline and liver diseases, such as NAFLD, fatty liver, and liver damage. Very few studies have been focused on animal models of ACC. We have added the following texts in the introduction part of the revised manuscript,
Current animal studies have focused on choline with liver[1,2]. There are few animal studies on the relationship between choline and ACC. Animal experiments are needed in the future.
- Miyachi Y, Akiyama K, Tsukuda Y, Kumrungsee T, Yanaka N (2021) Liver choline metabolism and gene expression in cho-line-deficient mice offspring differ with gender. Bioscience, biotechnology, and biochemistry 85:447-451.
- Mehedint MG, Zeisel SH (2013) Choline's role in maintaining liver function: new evidence for epigenetic mechanisms. Curr Opin Clin Nutr Metab Care 16:339-345.
- By searching the Medline database with the keyword "ABCG8", I see N=808 articles. I kindly ask you to supplement the references that are more important to you and expand the discussions with the relevant ones.
Response: We agree with the reviewer’s point that information related to ABCG8 should be supplemented in the manuscript.. Previous studies have demonstrated the role of ABCG8 in excretion of cholesterol[3, 4]. Cholestasis is one of the possible pathogenic factors of AAC. This strengthens the causal relationship between choline and AAC. In the revised manuscript, we have supplemented the following texts in the discussion section,
Previous studies have demonstrated the role of ABCG8 in excretion of cholesterol [3,4]. Taken together, choline metabolites may play a protective role against AAC by promoting cholesterol excretion and transport. The specific molecular mechanism still needs further animal experiments. Cholestasis is one of the possible pathogenic factors of AAC. This strengthens the causal relationship between choline and AAC.
- Coy DJ, Wooton-Kee CR, Yan B, Sabeva N, Su K, Graf G, Vore M (2010) ABCG5/ABCG8-independent biliary cholesterol ex-cretion in lactating rats. Am J Physiol Gastrointest Liver Physiol 299:G228-G235
- Yu X-H, Qian K, Jiang N, Zheng X-L, Cayabyab FS, Tang C-K (2014) ABCG5/ABCG8 in cholesterol excretion and athero-sclerosis. Clin Chim Acta 428:82-88